# An Energy Consumption Approach to Estimate Air Emission Reductions in Container Shipping

Ernest Czermański [1,*], Giuseppe T. Cirella [2] , Aneta Oniszczuk-Jastrząbek [1], Barbara Pawłowska [2] and Theo Notteboom [3,4,5,6]

1 Department of Maritime Transport and Seaborne Trade, Faculty of Economics, University of Gdansk, 81-824 Sopot, Poland; aneta.oniszczuk-jastrzabek@ug.edu.pl
2 Department of Transport Economics, Faculty of Economics, University of Gdansk, 81-824 Sopot, Poland; gt.cirella@ug.edu.pl (G.T.C.); barbara.pawlowska@ug.edu.pl (B.P.)
3 Center for Eurasian Maritime and Inland Logistics, China Institute of FTZ Supply Chain, Shanghai Maritime University, Shanghai 201306, China; theo.notteboom@ugent.be
4 Maritime Institute, Faculty of Law and Criminology, Gent University, B-9000 Gent, Belgium
5 Faculty of Business and Economics, University of Antwerp, 2000 Antwerp, Belgium
6 Antwerp Maritime Academy, 2000 Antwerp, Belgium
* Correspondence: ernest.czermanski@ug.edu.pl; Tel.: +48-502-241-414

**Abstract:** Container shipping is the largest producer of emissions within the maritime shipping industry. Hence, measures have been designed and implemented to reduce ship emission levels. IMO's MARPOL Annex VI, with its future plan of applying Tier III requirements, the Energy Efficiency Design Index for new ships, and the Ship Energy Efficiency Management Plan for all ships. To assist policy formulation and follow-up, this study applies an energy consumption approach to estimate container ship emissions. The volumes of sulphur oxide ($SO_x$), nitrous oxide ($NO_x$), particulate matter (PM), and carbon dioxide ($CO_2$) emitted from container ships are estimated using 2018 datasets on container shipping and average vessel speed records generated via AIS. Furthermore, the estimated reductions in $SO_x$, $NO_x$, PM, and $CO_2$ are mapped for 2020. The empirical analysis demonstrates that the energy consumption approach is a valuable method to estimate ongoing emission reductions on a continuous basis and to fill data gaps where needed, as the latest worldwide container shipping emissions records date back to 2015. The presented analysis supports early-stage detection of environmental impacts in container shipping and helps to determine in which areas the greatest potential for emission reductions can be found.

**Keywords:** container shipping; emissions; maritime transport; sustainable shipping; green shipping; IMO

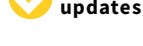



## 1. Introduction

The world's container shipping fleet (fully cellular) consisted of approximately 5600 vessels in 2018, representing only 8% of the total fleet tonnage in global shipping. However, in terms of sailing distance container ships are responsible for 17% of all maritime transport [1,2]. The container shipping fleet is diversely structured, both in terms of vessel size and cargo-carrying capacity, with eight principal sizes varying in numerical strength and capacity (Table 1). In December 2018, the container shipping fleet delivered a combined deadweight tonnage of 295,746,617 dwt. The average vessel was 224.2 m in length, 32.2 m in width, and had 11.2 m draught. The average technical speed for the entire fleet was 20.5 kn, while the average combined engine power was 28,089 kW, with the whole fleet amounting to 157,413,558 kW (see Supplementary Materials) [2]. The first four parameters impact a ship's energy requirement while the last two directly affect fuel consumption.

**Table 1.** Size and carrying capacity of the container shipping fleet by vessel type, 2018.

| | Size | | Carrying Capacity (TEU [1]) | |
|---|---|---|---|---|
| **Vessel Type** | **Total Vessels** | **Fleet Percentage** | **Average** | **Standard Deviation** |
| Containership-Small Feeder | 953 | 17.0 | 604.2 | 265.0 |
| Containership-Regional Feeder | 1393 | 24.9 | 1415.4 | 307.6 |
| Containership-Feedermax | 758 | 13.5 | 2530.7 | 241.9 |
| Containership-Sub-Panamax | 202 | 3.6 | 3368.3 | 246.8 |
| Container-Baby post-Panamax | 193 | 3.4 | 4362.8 | 526.7 |
| Containership-Panamax | 533 | 9.51 | 4508.3 | 335.0 |
| Containership-Post-Panamax | 925 | 16.5 | 7572.3 | 1382.4 |
| Containership-ULCS [2] | 647 | 11.6 | 14,543.0 | 3549.7 |
| Total | 5604 | 100.0 | 4426.3 | 4533.4 |

[1] twenty-foot equivalent unit. [2] ultra large container ship. Source: own compilation based on IHS Markit Portal [2].

According to data from 2015, container ships consumed 80 million metric tons of marine fuel, a figure amounting to 25% of total fuel consumption by all ships worldwide. To stress energy usage, container shipping represents 26% of the shipping industry's total energy intake [1]. As such, no other segment of maritime shipping can boast such extravagant figures [3]. Possible solutions to reduce fuel consumption include: (1) reducing ship speed, (2) installing auxiliary propulsion systems (e.g., kites), (3) streamlining the ship's hull (e.g., slippery layers on the submerged part of the hull) and (4) route optimization systems concerning navigation conditions, speed, heeling, and other voyage parameters to reduce fuel consumption throughout the voyage [4,5].

The International Maritime Organization (IMO) has estimated that the maritime shipping industry contributes 2.5 to 3.0% of annual human-produced carbon dioxide ($CO_2$) emissions in which the largest portion of that is derived from container shipping [1]. As such, container shipping is responsible for a significant part of the world's burning of fossil fuels and ocean pollution. Alongside $CO_2$, other greenhouse gas (GHG) emissions released by ships include $SO_x$, $NO_x$, and PM, which are highly toxic, create air pollution and cause acid rain (i.e., via irregular pH levels)—the health effects of pollutants on human health are often difficult to estimate as they depend on the substance as well as its concentration and exposure time. According to the World Health Organization (WHO) [6,7], approximately 1.3 million people die prematurely every year in cities as a result of urban air pollution. In Europe, according to the European Environmental Agency, there are 350,000 premature deaths due to over exposure to $PM_{2.5}$ and 20,000 premature deaths due to exposure to high $O_3$ concentrations [8,9]. The WHO has also shown that exposure to $PM_{2.5}$ accounts for 8% of lung cancer deaths, 5% of cardiovascular deaths, and 3% of respiratory infections worldwide [10]. It also indicates that an increased risk of morbidity and even mortality from respiratory diseases is associated with exposure to $NO_2$, also in concentrations below the limit values [11]. Another effect includes black carbon which reduces ice cover and overall absorption rates that create higher levels of heat due to positive radiative forces (e.g., climate change). The resulting short- and long-term effects on health are important reasons for limiting emission levels [12].

Due to the fact that container shipping is such a critical part of the global economy, ceasing container ship activity is not a feasible option [13]. Yet, it is essential that measures are implemented to first reduce and eventually halt emissions. To this end, some important steps have already been taken such as the judicial implementation of IMO's MARPOL Annex VI that binds limitations on the main air pollutants contained in ships' exhaust as well as further plans of implementing Tier III requirements, Energy Efficiency Design Index (EEDI) for new ships, and Ship Energy Efficiency Management Plan (SEEMP) for all ships [1,14].

To assist policy formulation and follow-up, this paper uses an energy consumption approach to determine air emissions generated by container shipping in 2018 and to

simulate reductions in emissions for 2020. Using the baseline information on vessel size and carrying capacity of the container shipping fleet, we piece together the estimated volume of air emissions from 2016 to 2020 using cross-referenced [2] data. The calculations are supported by a mapping analysis which portrays a visual illustration of the estimated potential for reducing shipping emissions. In term of regulatory measures, this paper should be aligned with the IMO's MARPOL Annex VI for $CO_2$ [1], Regulation 14 for $SO_x$ and PM [15] and Regulation 13 for $NO_x$ [16]. To better understand the shipping container industry's connection with emission levels the state of the art of technologies, as well as fuel options and other alternatives, are examined. The calculations from this paper bare some significance as they give a sneak peek into future estimation of emission levels generated by specific shipping routes and shipping lines as well as determine the amount of emissions per unit of a container ship's transport work. This will, in turn, help to determine average emissions per container-mile (i.e., unit of transport work). Moreover, the presented analysis supports early-stage detection of environmental impacts in container shipping and helps to determine in which areas the greatest potential for emission reductions can be found.

The paper is structured as follows. After a brief discussion on possible methodological approaches for the measurement of ship emissions, we introduce the building blocks of the energy consumption approach and provide more details on the data collection and design aspects of the research and the mapping of results. Section 4 elaborates on the empirical results with a focus on the worldwide energy requirements for the container shipping fleet. These outcomes are used to estimate the potential for reducing $SO_x$, $NO_x$, PM, and $CO_2$ emissions by comparing the situation for 2018 with future estimates under global cap conditions.

## 2. Methodological Approaches to the Measurement of Ship Emissions

There is a vast literature on the measurement of ship emissions including scientific work and regulation-related documents for IMO and others. One of the first seminal studies among the related IMO documents was in 2000, in which an international consortium, led by Marintek, delivered the "Study of Greenhouse Gas Emissions from Ships" which included an estimation of ship emissions for 1996 and an examination of emission reduction possibilities. The two main methodological approaches can be adopted to measure ship emissions: a fuel-based approach (top-down) and an activity-based (bottom-up) approach [17]. The first approach relies on marine fuel consumption data and fuel-related emission factors [18] and is particularly useful in the case no detailed information is available on ship movements. The activity-based approach method requires very detailed datasets on the technical characteristics and operations of individual vessels. Based on these datasets, emissions of an individual ship can be calculated and aggregated to obtain fleet emission estimates [19]. Some researchers have followed hybrid approaches by combining the top-down and bottom-up approaches. While the bottom-up methodology in principle generates more fine-meshed results, it requires vast amounts of reliable vessel and traffic data [20]. Additionally, the activity-based approach often relies on the use of average input parameters like engine load factors, time spent in port, fuel consumption rate, and emission factors which greatly depend on the ship characteristics (e.g., age, size, fuel type) and the (changing) market conditions [21].

Corbett and Köhler [22], Endresen et al. [23], Eyring et al. [24] were among the early studies presenting detailed methodologies for constructing fuel-based inventories of ship emissions mainly based on engine power and vessel activity data. These methods to measuring ship emissions have been refined and adapted in later studies. Using fuel consumption as a main input, Psaraftis and Kontovas [25] presented an analysis on emissions of the world fleet for major ship types, i.e., bulk carriers, crude oil tankers, container vessels, product or chemical carriers, liquefied natural gas (LNG) carriers, liquefied petroleum gas carriers, reefer vessels, Ro-Ro vessels, and general cargo ships. The literature review study of Nunes et al. [17] analyzed 26 papers published between 2010 and 2016 using the activity-based methodology to estimate ship emissions. Most of the authors allocating

emissions by ship type concluded that container ships were the main pollutant emitters. Cariou et al. [26] followed a mixed approach (bottom-up and top-down) by developing a model to estimate the total $CO_2$ emissions in container shipping per trade lane, using three building blocks: a trade-related port time module, a trade-related liner service module, and a vessel design speed fuel consumption module. The study reports an average $CO_2$ emission in 2016 of 58 g per TEU-km against 50 reported in a similar study by BSR Clean Cargo Working Group [27].

The fuel-based approach (top-down) and activity-based approach (bottom-up) have also been deployed to forecast ship emissions and associated environmental impacts. For example, Song and Shon [18] present a scenario-based analysis to predict ship emissions in the port of Busan for the medium and long-term (up to 2050), while Corbett et al. [28] produced forecasts for 2020, 2030, and 2050. Short-term vessel emission forecasts have been presented by Liu et al. [29] in the context of the Domestic Emission Control Areas in China. Liu and Duru [30] criticize the use of deterministic extrapolation and point estimates as a basis for emission forecasting and, therefore, propose a Bayesian algorithm approach which can generate a range of possible outcomes based on the probabilistic forecasting concept.

As mentioned earlier, this study applies an energy-consumption approach to determine air emissions generated by container shipping in 2018 and to simulate reductions in emissions for 2020. In view of determining the $CO_2$ emissions by global container shipping, this paper appends to the current worldwide $CO_2$ emission levels from shipping which dates back to 2007–2015 (i.e., Third IMO GHG Study from 2007 to 2012 and International Council on Clean Transportation's (ICCT) report from 2013 to 2015) (Table 2). The Fourth IMO GHG Study has been published in the last quarter of 2020 [31].

**Table 2.** Worldwide $CO_2$ emissions from international shipping as a total of global anthropogenic emissions, cargo carriage volume and transportation work performed, 2007–2015.

| | Third IMO GHG Study | | | | | | ICCT | | |
|---|---|---|---|---|---|---|---|---|---|
| | 2007 | 2008 | 2009 | 2010 | 2011 | 2012 | 2013 | 2014 | 2015 |
| Global $CO_2$ emissions [1] | 31,959 | 32,133 | 31,822 | 33,661 | 34,726 | 34,968 | 35,672 | 36,084 | 36,062 |
| Total shipping [1] | 1100 | 1135 | 977 | 914 | 1021 | 942 | 910 | 930 | 932 |
| International shipping [1] | 881 | 916 | 858 | 773 | 853 | 805 | 801 | 813 | 812 |
| Cabotage [1] | 133 | 139 | 75 | 83 | 110 | 87 | 73 | 78 | 78 |
| Fisheries [1] | 86 | 80 | 44 | 58 | 58 | 51 | 36 | 39 | 42 |
| Share of shipping in global $CO_2$ emissions | 3.4% | 3.5% | 3.1% | 2.7% | 2.9% | 2.6% | 2.5% | 2.6% | 2.6% |
| Cargo carriage [2] | 8034 | 8229 | 7858 | 8409 | 8785 | 9197 | 9514 | 9843 | 10,024 |
| Transportation work [3] | 41,177 | 42,167 | 40,308 | 44,603 | 46,834 | 48,937 | 50,490 | 52,627 | 53,476 |
| Correlation factor ($E^T$) | 0.277 | 0.269 | 0.242 | 0.205 | 0.218 | 0.193 | 0.180 | 0.177 | 0.174 |

[1] million metric tons/year; [2] volume calculation is done using the STEAM3 model sourced from [3]; [3] million metric tons/Nm; correlation coefficient, i.e., $E^T$, represents the ratio between the amount of $CO_2$ emissions from shipping in tons of $CO_2$ and amount of annual transportation work carried out by the world fleet in tons of $CO_2$. Source: authors own elaboration based on Comer et al. [32] and statistical data from UNCTAD/RMT/2018 [33].

## 3. Methodology

### 3.1. Methodological Scope of the Energy Consumption Approach

In this paper, we present an energy consumption approach to the measurement of ship emissions of the container shipping fleet, i.e., a conceptual alternative energy approach, to measure worldwide energy requirements for container shipping. This approach considers the size of the fleet and parameters related to the required energy to operate this fleet. As

indicated earlier, the application of the energy consumption approach can be utilized as a tool-based method when limited or no data records on detailed container ship movements are made available. It allows for the estimation of ship fuel consumption and emitted air emissions in the course of ship operations. A compilation of relevant air emissions data has been developed based on the various stages during the round voyage of a ship, i.e., total berth time, maneuvering within the port, entry into port, anchoring and—most importantly—the actual voyage at sea. This approach enables for a more thorough analysis of the environmental impact of shipping and helps to determine in which areas action can be taken to reduce air emissions and where the greatest potential for such reductions lie.

### 3.2. Data Collection, Design, and Variables

A database on container ships was created specifically for this research, with data for December 2018. The base dataset was purchased from the IHS [2] Maritime Portal. The data focuses on parameters crucial for estimating emissions and correlating the amount of fuel consumed by ships. Fuel consumption has been determined based on an assigned value to each ship in relation to its cargo-carrying capacity. Using this baseline method, a container ship with a carrying capacity of 1000 TEU, tested within the speed range of 10–17 kn, corresponds to Froude numbers ranging between 0.16 to 0.28 (Figure 1) [34–36]. It should be emphasized that the shape of the identified graph is cross-compatible with larger vessels without incurring a risk of methodological error. The calculation shows a steep increase in fuel consumption as speed increases. Systematically, at 10 kn the daily fuel consumption reaches 4.8 metric tons, at 15 kn (i.e., a 50% increase) it is 14.4 metric tons (i.e., 300% more), while at 17 kn (i.e., 13% up from 15 kn) the daily fuel consumption reaches 33.6 metric tons (i.e., 233% on top of the preceding amount). These calculations indirectly point to a strong increase in the levels of pollutants into the air as the ship speed increases.

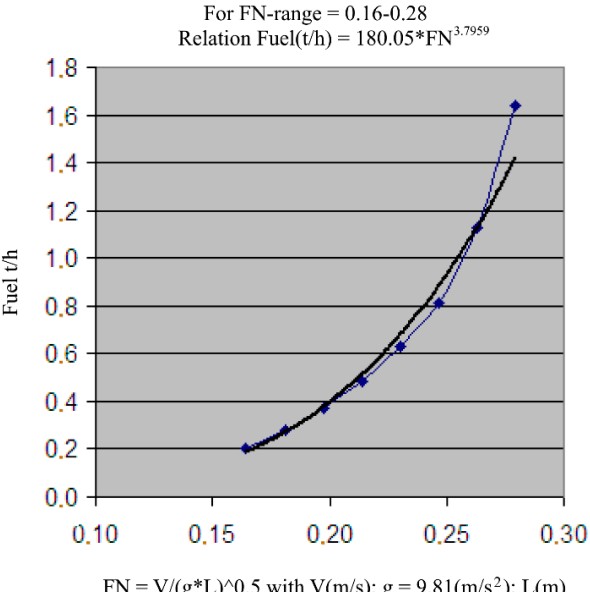

For FN-range = 0.16-0.28
Relation Fuel(t/h) = $180.05 \ast FN^{3.7959}$

FN = $V/(g \ast L)$^0.5 with V(m/s); g = 9.81(m/s$^2$); L(m)

**Figure 1.** An example function illustrating fuel consumption relative to the ship's speed expressed as a Froude number.

To translate the results from energy use to emissions, the calculated fuel consumption figures have been correlated with emission level indices for each of the specific fuel types. The most widespread indicators link air emissions to energy density by using the following fuel types: heavy fuel oil (HFO) (i.e., 40.0 MJ/kg of fuel), marine diesel oil (MDO) (i.e., 42.7 MJ/kg) and LNG (i.e., 50.0 MJ/kg). The indices were converted in such a manner as to translate consumption of a given fuel (i.e., in metric tons) into emissions produced in the process of consumption (i.e., in metric tons of $CO_2$ and kilograms for other pollutants).

The results are presented in Table 3. The calculations identify the total air emissions for basic pollutant compounds derived from container shipping.

**Table 3.** Emissivity indices for selected marine fuels.

| Item | Fuel | [kg/t of Fuel] | | | | Energy Density Relative to MDO (%) |
|---|---|---|---|---|---|---|
| | | $CO_2$ | $SO_X$ | $NO_X$ | $PM_{2.5}$ | |
| 1 | MDO 0.5% | 3206.00 | 10.50 | 50.50 | 2.30 | 100 |
| 2 | HFO 1.5% | 3114.00 | 31.50 | 51.00 | 3.40 | 94 |
| 3 | HFO 2% | 3114.00 | 42.00 | 51.00 | 3.40 | 94 |
| 4 | HFO 3.5% | 3114.00 | 71.50 | 51.00 | 3.40 | 94 |
| 5 | LSHFO 0.5% | 3151.00 | 10.50 | 51.00 | 2.30 | 94 |
| 6 | LSMGO 0.1% | 3151.00 | 2.10 | 50.50 | 2.30 | 100 |
| 7 | LNG | 2750.00 | <0.02 | 8.40 | 0.02 | 1170 |
| 8 | Methanol | 1375.00 | 0.00 | 26.10 | 0.02 | 46.8 |
| 9 | HFO + SCRUBBER + SCR [1] | 3176.28 | 0.84 | 7.65 | 0.51 | 94 |

[1] selective catalytic reduction. Source: authors own elaboration based on the assumptions of the Med Atlantic Ecobonus (MAE) Project, MAE External Cost Calculator Tool [37].

ICCT was consulted to determine the structure of marine fuels used in container shipping [32]. The predominant fuel used for the dataset is HFO whose percentage amounts to 93%, with the remaining 7% referring to distillates (Figure 2).

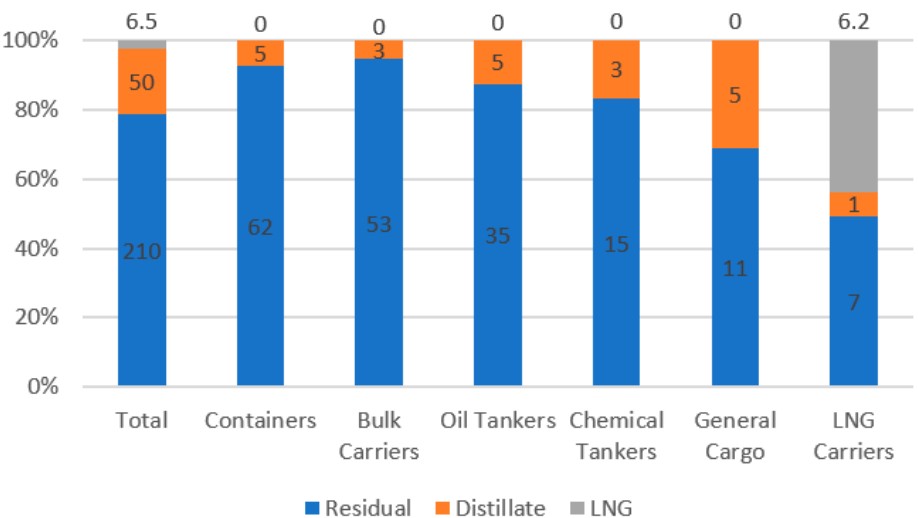

**Figure 2.** Classification by type of marine fuel according to the shipping sector, adapted from Comer et al. [32].

LNG accounts for a marginal share in the analyzed dataset, considering that the entire fleet had only four vessels powered by this type of fuel. Such a low share of LNG, within the overall fuel consumption figure, calls for a thorough analysis in order to explore the possibility of modernizing the fleet with a view to reducing overall emission levels.

*3.3. Mapping*

The mapping analysis portrays a visual illustration of the estimated potential for reducing shipping emissions. Then, two software packages were used for Geographic Information Systems (GIS) imagery and overlay work, i.e., software Ersi ArcGIS (version 10.7) and software Paint.Net (version 4.1.6). The GIS mapping calculations used the mean total for emission levels (i.e., $SO_x$, $NO_x$, PM, and $CO_2$) to deduct results from Equation (1).

$$R_r = delta(X_v) \: / \: X(V_{n0}) \tag{1}$$

where: $R_r$ = reduction rate (mean), $delta(X_v)$ = difference in existing emission level $(X_o)$ versus predicted emission level $(X_1)$ by specific speed "$V$", "$n$" = speed between 14 and 18 kn, $X(V_{n0})$ = existing emission volume for specific "$n$" speed "$V$".

We apply the energy consumption approach to calculate the consumed energy (i.e., fuel) from container ships on a global scale. Due to a lack of records, automatic identification system (AIS) datasets (i.e., for all container ships worldwide) were single-handedly calculated to obtain real fuel consumption levels [2]. This method has not been used for the whole of the container shipping sector and provides first account and valuable intermediate results before the Fourth IMO GHG Study is published in the last quarter of 2020 [31].

### 3.4. Empirical Results

In a first step, the results focus on the worldwide energy requirements for the container shipping fleet. The fuel consumption type is determined via fuel use patterns and maximum continuous rating (MCR). Then, we identify the estimated potential for reducing $SO_x$, $NO_x$, PM, and $CO_2$ in four subsequent subsections. For each of these subsections emissivity is generated in 2018, future estimates are presented (i.e., under global capped conditions) and mapped results for 2020 are presented using the energy consumption approach.

### 3.4.1. Energy Requirements in Container Shipping

The world's container shipping fleet, which included some 5600 vessels in 2018, represents 149,483 MW of power available from auxiliary engines and power generators, translating into an energy requirement of 179 GW. When assuming operations at maximum performance (i.e., MCR = 1.0) annually (i.e., per 365 days) and an average requirement for HFO of 180 g/1kWh, the maximum fuel requirement is estimated to be 282,000,000 metric tons [1]. If we consider the average sailing time only (i.e., circa 250 days), the figure drops to around 193,000,000 metric tons. This is the actual fuel consumption figure for the entire global container shipping fleet. According to data from 2012, fuel consumption stood at 276,000,000 metric tons, of which 195,000,000 represented HFO and 81,000,000 MGO, respectively [1–3,19,38].

In order to calculate the annual emissions produced by container shipping, the average power output use is reduced to MCR = 0.85. This is, in fact, the power output level adopted for this research in which periodic change could occur depending on a ship owner's strategy and current transport needs. Generally, however, this power output factor is valid, as vessels do not operate at maximum power due to the excessive fuel consumption and elevated risk of emergency this would bring. According to Comer et al. [32], the volume of fuel consumed in 2015 in the container shipping industry amounted to 66,860,000 metric tons, representing 25% of the combined use of all types of fuel in the world [39,40]. As noted, 93% of that figure accounts for HFO residual fuel (i.e., 30% of global use) and 7% distillates (i.e., 10%). In the case of HFO fuel usage, container shipping is the largest consumer in the entire shipping industry, while distillates are predominately used in ferry and ro-ro shipping. LNG consumption was measured at around 3000 metric tons which is a negligible value compared to the overall 6,200,000 metric tons consumed by gas carriers.

Fuel consumption was analyzed by type of shipping to determine fuel consumption patterns at the various stages of a voyage. A voyage consists of four stages: (1) moored at berth for cargo handling operations (including idle time), (2) maneuvering (i.e., at the port), (3) anchoring (i.e., before entering the port and while awaiting further instructions) and (4) time at sea (i.e., sailing time) [14,32]. Container ships use a mere 8% of fuel outside of the time spent at sea, as shown in Figure 3. This means the primary focus area for measures to reduce air emission levels should be when ships are sailing at sea. The ship owner usually has only partial control over the length of the anchoring stage, as it depends on congestion within the port and access-preventing weather conditions. Coordination between route optimization systems used by the ship operator and vessel traffic management systems (VTMS) used to manage port access routes can help to avoid any waiting time".

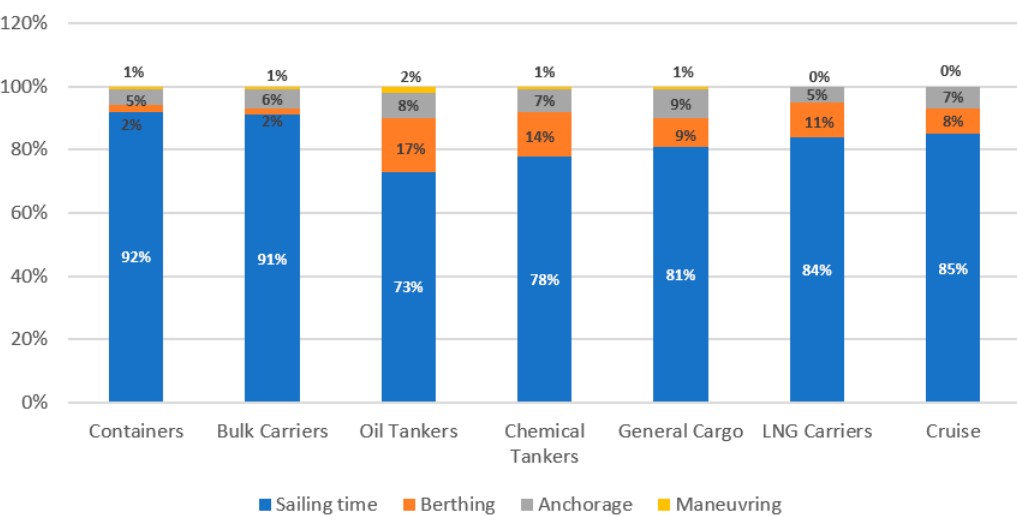

**Figure 3.** Energy consumed by ships according to the stage of voyage and type of fleet, adapted from Comer et al. [32].

Remarkably, container shipping is characterized by fuel use disproportions across the various stages of the sea voyage. According to Comer et al.'s [32] survey, as shown in Figure 3, cargo handling time in port is kept to a minimum as a result of the pressure exerted by shipping lines on terminal operators to reach a high terminal productivity in terms of the amount of cargo loaded and discharged per time unit.

To determine the fuel volume consumed by container ships in 2018, the technical parameters of ships, relating to engine power, cargo-carrying capacity in TEU and length were examined. Table 4 presents the calculated totals. The volume of fuel consumed in the container shipping industry was estimated at 117,800,000 metric tons in 2018. It was assumed for the purpose of the calculations that this figure is a theoretical value corresponding to 17 kn, i.e., a speed level approximating the average commercial speed reported by Marine Traffic [41] and confirmed by the authors in trial measurements executed in 2018.

**Table 4.** Calculated fuel consumption figures for container ships in 2018.

| V [kn] | $C_{MGO}$ [t] | $C_{HFO}$ [t] | $\sum C_{(MGO+HFO)}$ [t] |
|---|---|---|---|
| 14 | 6,821,449 | 90,627,824 | 97,449,273 |
| 15 | 7,333,524 | 97,032,536 | 104,366,060 |
| 16 | 7,784,678 | 103,425,006 | 111,209,684 |
| 17 | 8,247,186 | 109,569,756 | 117,816,942 |
| 18 | 9,852,961 | 130,903,629 | 140,756,590 |

Source: own calculations based on data from IHS Markit Portal [2].

To fully illustrate the relationship between fuel consumption and speed, Table 4 sets out theoretical values for other speeds within the range of 14–18 kn. The obtained results will be used to calculate emissivity to air for the most important compounds such as $SO_x$, $NO_x$, PM, and $CO_2$.

### 3.4.2. Estimated Potential for Reducing $SO_x$ Emissions

Container shipping stands out as using a very large share of HFO-type residual fuel with various levels of leftover sulphur content. After establishing Sulphur Emission Control Area (SECA) zones, the fuel used in ship propulsion systems was 3.5%. Prior to 1 January 2015, low-sulphur 1.5% HFO fuel was used. Afterwards, ship owners switched to MGO compliant with the 0.1% requirement in which the use of scrubbers is minimal and no conversions to LNG were noticed.

As a result, it would be correct to say that the previously large demand for 1.5% HFO fuel has disappeared or declined considerably. A direct consequence of this is the reduced emissivity of sulphur oxides. According to calculations made from our approach,

7,850,000 metric tons of sulphur compounds were emitted in 2018 in container shipping alone. This is an approximate number that can be verified by tracking ships in motion at sea with the AIS and locating their position relative to the SECA zone, which can be challenging for future research projects [20,42]. To illustrate the changes that can occur as a result of variations in ship speed, Table 5 shows the calculated $SO_x$ emissions for the speed range of 14 to 18 kn. It is evident a clear positive, though slightly regressive, correlation depicts $SO_x$ emissivity rises, although at decreasing rates, in step with increasing speed.

**Table 5.** Estimated figures for $SO_x$ emissivity generated by container shipping in 2018.

| V [kn] | $SO_x$ MGO [t] | $SO_x$ HFO [t] | $\sum SO_x$ [t] |
|--------|----------------|----------------|-----------------|
| 14 | 14,325 | 6,479,889 | 6,494,214 |
| 15 | 15,400 | 6,937,826 | 6,953,227 |
| 16 | 16,348 | 7,394,888 | 7,411,236 |
| 17 | 17,319 | 7,834,238 | 7,851,557 |
| 18 | 20,691 | 9,359,609 | 9,380,301 |

Source: own calculations based on data from IHS Markit Portal [2].

These calculations can be supplemented with an attempt to determine $SO_x$ emissions following the entry into force of the so-called global cap in 2020. The calculations used emissivity coefficients for 0.5% fuel which has become the most widespread standard in the face of the existing cap. It was concluded based on an analysis of how many ships on order, whose delivery date falls after 2019, were equipped with scrubbers. The total number of these ships will reach 190 after 2020. Table 6 illustrates their percentage within the entire existing fleet and the number of ordered ships in total.

**Table 6.** Ordered container ships to be delivered in 2019–2022.

| Year | Number of Scrubber-Equipped Ships | Total | Scrubber-Equipped Ships (%) | Total Fleet (%) |
|------|-----------------------------------|-------|-----------------------------|-----------------|
| 2019 | 26 | 160 | 16.25 | 2.24 |
| 2020 | 53 | 155 | 34.20 | 3.13 |
| 2021 | 16 | 39 | 41.00 | 3.40 |
| 2022 | 0 | 3 | 0.00 | 3.39 |
| Total | 95 | 403 | 23.60 | – |
| Container ship fleet in operations in 2018 | | | | |
| Until 2018 | 95 | 5238 | 0.17 | 1.71 |

Source: authors own elaboration based on IHS Markit Portal [2].

There were 24 LNG-powered ships on order in late 2018 (i.e., this total does not include 2 ships whose delivery, scheduled for 2018, could not be confirmed). The data indicates LNG is not an alternative for 0.5% HFO since emissions were still found to exist at significant levels; however, other assumptions remained as they were. Unsurprisingly, $SO_x$ emissions will face a drastic reduction by more than 85%, translating quantitatively to 6,600,000 metric tons for HFO alone (Table 7). Using GIS, annual $SO_x$ emissions for 2020 visually illustrate a change from high to low levels with the introduction of the energy consumption approach (Figure 4).

A world-scale limit of sulphur content set at 0.1% requires either a full transition to 0.1% MGO fuel, installation of scrubbers, or using another solution (i.e., the latter is not included in the calculations). Such a development allows for an even greater reduction of $SO_x$ which has been linked as the precursor to acid rain and atmospheric particulates [43]. In other words, a global limit of 0.1% (i.e., implemented via worldwide SECA regulation) minimizes sulphur emissions which amount to 230,000 metric tons annually. This signals a significant step in maneuvering toward zero-emissivity of $SO_x$ in container shipping.

**Table 7.** Estimated amount of SO$_x$ emissivity in container shipping under global cap conditions (i.e., 0.5% HFO fuel) for 2020.

| V [kn] | SO$_x$ MGO [t] | SO$_x$ HFO [t] | $\sum$ SO$_x$ [t] |
|---|---|---|---|
| 14 | 14,325 | 951,592 | 965,917 |
| 15 | 15,400 | 1,018,842 | 1,034,2420 |
| 16 | 16,348 | 1,085,963 | 1,102,310 |
| 17 | 17,319 | 1,150,482 | 1,167,801 |
| 18 | 20,691 | 1,374,488 | 1,395,179 |

Source: based on extrapolated data from IHS Markit Portal [2].

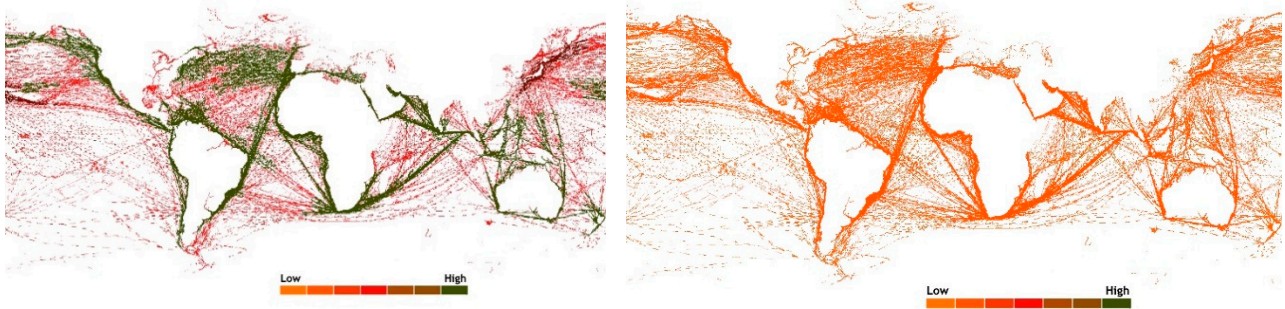

**Figure 4.** Annual global container shipping SO$_x$ emission levels for 2020, (left) base situation for 2018 and (right) estimates using the energy consumption approach, based on extrapolated data from IHS Markit Portal [2].

### 3.4.3. Estimated Potential for Reducing NO$_x$ Emissions

Currently, the use of NO$_x$ emissions, derived from fuels used in container shipping, does not have any major reduction plans with exception of the implementation of the Tier III limits. According to results obtained for our approach, NO$_x$ emissions for 2018 amounted to 6,000,000 metric tons (Table 8). The results indicate the determined emissions would grow in step with increasing speed, however, at a progressively slower rate.

**Table 8.** Estimated amount of NO$_x$ emissions generated by container shipping in 2018.

| V [kn] | NO$_x$ MGO [t] | NO$_x$ HFO [t] | $\sum$ NO$_x$ [t] |
|---|---|---|---|
| 14 | 344,483 | 4,622,019 | 4,966,502 |
| 15 | 370,343 | 4,948,659 | 5,319,002 |
| 16 | 393,126 | 5,274,675 | 5,667,801 |
| 17 | 416,483 | 5,588,058 | 6,004,540 |
| 18 | 497,574 | 6,676,085 | 7,173,660 |

Source: based on extrapolated data from IHS Markit Portal [2].

The limit for sulphur content in marine fuel at 0.5% since 2020 does not affect the volume of NO$_x$ emissions. Our approach has shown that substituting 0.5% HFO fuel for 3.5% HFO has no effect on NO$_x$ emissivity. However, crucial changes will come in the wake of the full implementation of the Tier III conditions following 2021. This approach uses NO$_x$ emissions reduction parameters set for SCR installations at 7.65 g per metric ton of fuel, irrespective of the fuel type. SCR purifies ship engine exhaust gases no matter where it is installed. Assuming that the Tier III would extend to all sea areas, the resulting NO$_x$ emissions reduction could reach 79.1%, translating to a reduction of 4,749,849 metric tons annually compared with the existing emissions (Table 9). For 2020, the GIS analysis for NO$_x$ emissions show a change from high to low levels with the introduction of the energy consumption approach (Figure 5).

**Table 9.** Estimated amount of NO$_x$ emissions in container shipping under Tier III conditions (i.e., with SCR used) for 2021.

| V [kn] | NO$_x$ MGO [t] | NO$_x$ HFO [t] | $\sum$ NO$_x$ [t] |
|--------|----------------|----------------|-------------------|
| 14 | 52,184 | 693,303 | 1,037,786 |
| 15 | 56,101 | 742,299 | 1,112,642 |
| 16 | 59,553 | 791,201 | 1,184,327 |
| 17 | 63,091 | 838,209 | 1,254,691 |
| 18 | 75,375 | 1,001,413 | 1,498,987 |

Source: own calculation based on data from IHS Markit Portal [2].

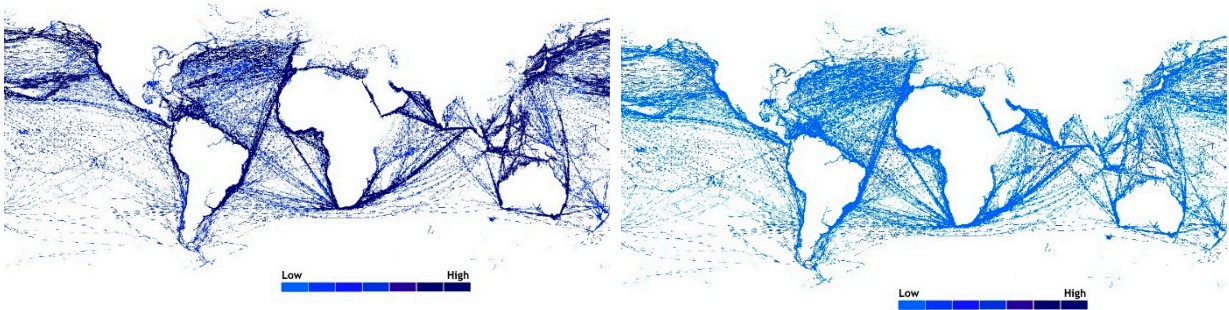

**Figure 5.** Annual global container shipping NO$_x$ emission levels for 2020, (left) base situation for 2018 and (right) estimates using the energy consumption approach, based on extrapolated data from IHS Markit Portal [2].

As such, it is worth noting that possible efforts to reduce ship-derived NO$_x$ emissions end there, as no other traditional fuel generates less emissions than this compound. Only possible alternatives beyond nitrogen oxide fuel would be the introduction of renewable sources such as hydrogen or electrically powered propulsion systems in which the sourcing process of these fuels (i.e., including electricity generated batteries or liquid hydrogen production) would equate to zero-emissivity [44].

### 3.4.4. Estimated Potential for Reducing PM Emissions

In quantitative terms, ship-derived particulate emissions seem to put a smaller strain on the environment. As shown in Table 10, these emissions in container shipping could amount to around 391.5 metric tons in 2018. However, we should keep in mind the strongly negative effect of PM on human health and its contribution to icecap melting around the world. Just as with other compounds, the size of PM emissions is growing in step with the increase in ship speed and fuel consumption. Additionally, just as in the previous cases, the increases are degressive.

**Table 10.** Estimated amount of PM emissions generated by container shipping in 2018.

| V [kn] | PM MGO [t] | PM HFO [t] | $\sum$ PM [t] |
|--------|------------|------------|---------------|
| 14 | 15,689 | 308,135 | 323,824 |
| 15 | 16,867 | 329,911 | 346,778 |
| 16 | 17,905 | 351,645 | 369,550 |
| 17 | 18,968 | 372,537 | 391,506 |
| 18 | 22,662 | 445,072 | 467,734 |

Source: own elaboration based on data from IHS Markit Portal [2].

As the kind of fuel used by ships has a considerable effect on the volume of PM emissions, the calculation adopted a set global limit of 0.5% for 2020. The results of the calculations are presented in Table 11, suggesting that the transition to 0.5% HFO will enhance the reduction in PM emissions by 30.8%. In quantitative terms, under reference conditions, this would result in a reduction of more than 120,000 metric tons. The GIS

analysis for PM emission levels in 2020 illustrate the change from high to medium with the introduction of the energy consumption approach (Figure 6).

**Table 11.** Estimated amount of PM emissions in container shipping under global cap conditions (i.e., 0.5% HFO fuel) for 2020.

| V [kn] | PM MGO [t] | PM HFO [t] | ∑ PM [t] |
|---|---|---|---|
| 14 | 15,689 | 208,444 | 224,133 |
| 15 | 16,867 | 223,175 | 240,042 |
| 16 | 17,905 | 237,877 | 255,782 |
| 17 | 18,968 | 252,010 | 270,979 |
| 18 | 22,662 | 301,078 | 323,740 |

Source: own calculations based on data from IHS Markit Portal [2].

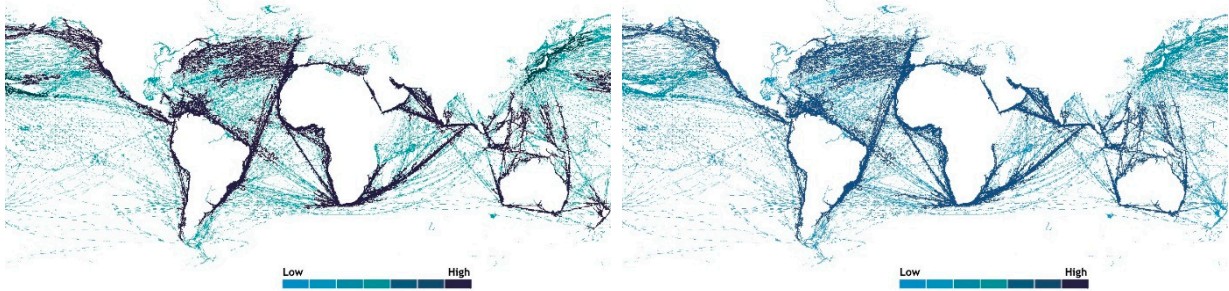

**Figure 6.** Annual global container shipping PM emission levels for 2020, (left) base situation for 2018 and (right) estimates using the energy consumption approach, based on extrapolated data from IHS Markit Portal [2].

It should be noted that further reductions in PM emissions are possible. Firstly, a shift towards LNG- or methanol-powered engines will bring about a radical decrease in these emissions to 2000 metric tons annually, a desirable step towards zero-emissivity, as well as, as previously discussed, the switch and implementation of hydrogen fuel or electrical propulsion systems.

### 3.4.5. Estimated Potential for Reducing $CO_2$ Emissions

The most thorough survey of $CO_2$ emissions is the 2017 ICCT report based on 2015 data. Besides $SO_x$, $NO_x$, PM and $CO_2$, the analysis also looked at CO, BC, $N_2O$ and $CH_4$ emissions. The analysis of $CO_2$ emissions in maritime shipping shows that its volume in 2015 came close to 1,000,000,000 metric tons. The container shipping industry, which operates a relatively small number of ships, was responsible for the largest portion of that figure amounting to 23%, or a 4-point percent more than the bulk carrier fleet which is twice as large in number [32]. Representing the most important area of ship-derived emissions, emitted $CO_2$ was calculated using our approach in Table 12. The relational importance of $CO_2$ emissions is due to it being the largest in volume, amounting to 367,200,000 metric tons annually, and primarily centered on formalizing emissions regulations over the next three decades. The main factor affecting the volume of $CO_2$ emissions is the emissivity coefficient defining a ratio of 3.1 metric tons of $CO_2$ emissions per 1 metric ton of fuel consumed. Another concern is the fact that MGO, though more environmentally friendly in terms of sulphur and PM emissions, generates more $CO_2$ emissions than poor-quality residual fuel (i.e., these increases are shown in quantitative terms in Table 13). Incrementally, the data indicates this type of emission may reach 371,200,000 metric tons annually, exceeding the 2018 limit by more than 4,000,000 metric tons (Table 14), amounting to 1.1% of total $CO_2$ emissions. Our research indicates that setting limits to sulphur oxide emissions is counterproductive and leads to a corresponding increase in $CO_2$ emissions. In this context it is worth to recommend paying special attention always when new regulations and

restrictions will be issued. The analyzed case shows that limiting one aspect can have a negative impact in another. A holistic approach is therefore, at most, essential.

**Table 12.** Estimated amount of $CO_2$ emissions generated by container shipping in 2018.

| V [kn] | $CO_2$ MGO [t] | $CO_2$ HFO [t] | $\sum CO_2$ [t] |
|---|---|---|---|
| 14 | 21,494,386 | 282,215,044 | 303,709,430 |
| 15 | 23,107,934 | 302,159,317 | 325,267,251 |
| 16 | 24,529,520 | 322,065,469 | 346,594,989 |
| 17 | 25,986,883 | 341,200,220 | 367,187,103 |
| 18 | 31,046,680 | 407,633,901 | 438,680,581 |

Source: own calculations based on data from IHS Markit Portal [2].

**Table 13.** Estimated amount of $CO_2$ emissions in container shipping under global cap conditions (i.e., 0.5% HFO fuel) for 2020.

| V [kn] | $CO_2$ MGO [t] | $CO_2$ HFO [t] | $\sum CO_2$ [t] |
|---|---|---|---|
| 14 | 21,494,386 | 285,568,273 | 307,062,659 |
| 15 | 23,107,934 | 305,749,521 | 328,857,455 |
| 16 | 24,529,520 | 325,892,194 | 350,421,714 |
| 17 | 25,986,883 | 345,254,301 | 371,241,184 |
| 18 | 31,046,680 | 412,477,335 | 443,524,015 |

Source: own calculations based on data from IHS Markit Portal [2].

**Table 14.** Changes in estimated amounts of $CO_2$ emissions generated by container shipping under global cap conditions as compared to 2018.

| V [kn] | $CO_2$ MGO [t] | $CO_2$ HFO [t] | $\sum CO_2$ [t] |
|---|---|---|---|
| 14 | 0.0 | −3,353,229 | −3,353,229 |
| 15 | 0.0 | −3,590,204 | −3,590,204 |
| 16 | 0.0 | −3,826,725 | −3,826,725 |
| 17 | 0.0 | −4,054,081 | −4,054,081 |
| 18 | 0.0 | −4,843,434 | −4,843,434 |

Source: own calculations based on data from IHS Markit Portal [2].

## 4. Discussion

We have examined the 2018 global container shipping fleet using the energy consumption approach with predictions for 2020. The findings are useful especially during interim breaks when the IMO or other international bodies have not released up to date and accurate information on emission levels in container shipping. A number of energy-related solutions to energy efficiency are worth noting. First, possible alternative propulsion via liquefied hydrogen fuels or electricity can prove to be very efficient [45–50]. A middle-of-the-road solution may also include switching to methanol. However, this is not practical due to limited output of this fuel globally as well as considerable costs related in converting propulsion systems. Second, viability may be supplied by way of new ship propulsion technology that will be introduced in response to the EEDI increasing energy efficiency standards. This can be seen clearly in the analysis of $CO_2$ emitted by container ships expressed in grams per ton-mile and according to vessel age, a correlation illustrated by the dispersed method in Figure 7. It illustrates dominant shifts towards larger TEU capacity and analyses concerned mainly with new ships (i.e., up to five years old). It is in this particular group of ships that the lowest rate of $CO_2$ emissions per unit of transport work exists. The future for air emissions will directly correspond with the EEDI standards which will operate in five-year cycles to reduce $CO_2$ emissions on new ships only, as well as the SEEMP for all ships. The EEDI is likely to become the most reliable driving factor behind shipowners' decisions, especially those operating container ships, to reduce emissions. Following 2025, these shipowners may be subject to pressure from technological

restrictions. This follows the fact that traditional fuels and propulsion systems, including LNG, equipped with additional installations will no longer be enough to comply with the currently applicable limit on $CO_2$ emissions. The propulsion systems acceptable will be the ones powered by hydrogen, electricity, and methanol obtained from renewable sources.

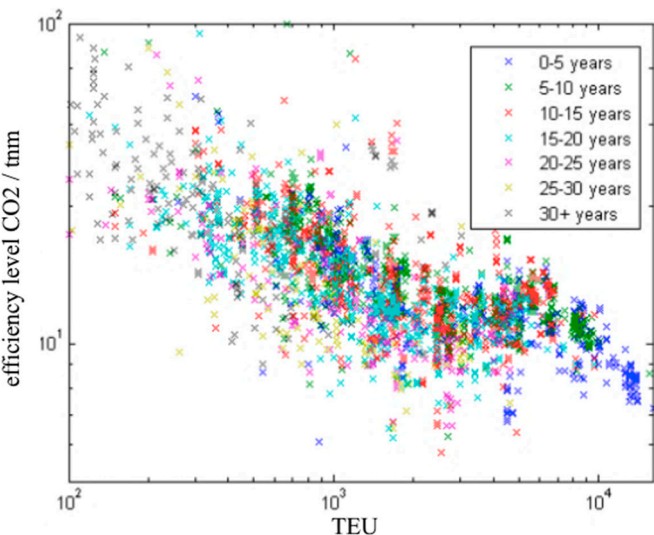

**Figure 7.** Container ship size correlated with emitted $CO_2$ (i.e., efficiency level gCO$_2$/tnm) per unit of transport work against TEU capacity, according to vessel age, adapted from Smith et al. [51].

In terms of the calculations relating to ship speeds, the used speed range covers 14 to 18 kn. If the speed of 17 kn is adopted as a reference point, we find the total annual fuel consumption in 2018 at 117,800,000 metric tons. The emission levels for this equates to over 7,850,000 metric tons of $SO_x$, 6,000,000 metric tons of $NO_x$, 391,000 metric tons of PM and 367,200,000 metric tons of $CO_2$. These figures illustrate that air emissions generated by global container shipping are significant. The latest data utilized from 2015 on $CO_2$ emissions amounts to just below 1,000,000,000 metric tons. The authors have determined the potential extent of reducing principal pollutants and air emissions after 2020 (i.e., in conjunction with global cap requisites) as a standard in upholding the United Nations Sustainable Development Goal 12 that ensures sustainable, responsible consumption and production [52]. Notably, the total emissions place container shipping as one of the top ranked emission activities (i.e., after electricity generation and industrial production). A further analysis of the data shows that the expected reduction in $SO_x$ may reach an elevated 85%, while for PM a 33% reduction is estimated. Following 2021, the expected reduction in $NO_x$ (i.e., when the Tier III limit is fully implemented) has been estimated at around 80%. As for $CO_2$ emissions, both the discussed limits will paradoxically swell emissions by around 4,000,000 metric tons annually as a result of transition to LSMGO fuel and increased demand for energy from scrubbers and SCR.

We should also consider a scenario of amending IMO regulations relating to EEDI by considering any innovative propulsion systems that may be developed, which will not only comply with the latest emissions limits but also—and most importantly—be affordable for ship owners. To best piece together best practices, the IMO's SEEMP measures and controls harmful air emissions from ships. Its implementation in coordination with EEDI tools can properly check ship pollution and act as an international baseline for managing ship energy efficiency, as well as the habits and know-how of the crew. Advanced features for implementation include new ship design and modification, as well as existing and older ship upgrading. Example amendments consist of innovation in hull shape, improved paints and antifouling, and various other techniques for wake equalization and flow separation. In terms of existing ships, to ensure energy efficiency is maximized they should maintain: (1) generators within an energy power management system in which they are run on as

high load as possible when burning heavy fuel; (2) the stoichiometric ratio (i.e., air-fuel mixture); (3) high priority maintenance of fuel system to safeguard correct fuel viscosity before injectors, scavenge pressure control, boiler and economizer and viscotherm; and (4) performance of the main engine. As such, the combination of planning, implementation, monitoring, self-evaluation and improvement, speed optimization, weather routing, hull monitoring and maintenance, efficient cargo operation, and electric power management all contribute to optimizing best practices and overall ship efficiency. Ship efficiency enacts an important part of sustainable thinking in combination with fuel air emission data and research.

## 5. Conclusions

This study presented an energy consumption approach to determine the annual rate of air emissions generated by container shipping. Data energy projections have been updated for 2020 taking into account changes in energy choices in the wake of the implementation of the global cap in early 2020. The estimation of the fuel consumption level depends directly on ship speed which varies considerably throughout a ship's voyage. This issue is paramount in the reliability of the presented approach since emission level measures are at its highest at sea (i.e., not waiting at the quay, maneuvering within port, entering into port or anchoring). As such, it is necessary to adopt average values for the various fuels (i.e., HFO, MGO, and—marginally—LNG). This approach adds to the knowledge base for understanding on how worldwide container shipping can be optimized for such reductions. In line with IMO's initiatives in implementing MARPOL Annex VI (i.e., the limitation on air pollutants from ship exhaust) as well as impending Tier III conditions and the EEDI requirements for new ships. It is paramount to develop sustainable, reliable and state-of-the-art transport systems based on quality and resilient design. Further research would be beneficial if the IMO or any other international organization fails to update the latest worldwide container shipping records dating back to 2015.

The demand for container shipping is likely to increase in the long term, despite the negative effect on demand of major shocks such as the financial-economic crisis of 2009 and the (ongoing) COVID-19 crisis. The container box has become an essential part of the world economy and associated global supply chains [53]. To secure its future 'license to operate', the container shipping industry is challenged to measure emission levels and plan for their reduction to aid in decreasing air pollution and related side effects. The energy consumption approach applied to container shipping provides estimation-based figures for a sound understanding of current emission levels. To further develop an exact real-time record, sensor-based devices can be implemented and cross-integrated into the AIS. This practice would add an additional level of precision which could be optimized from a centralized hub or online source. From an analytical viewpoint, early-stage detection and continuous monitoring can help to better understand environmental impacts and to determine what areas offer the greatest potential for emission level reductions. The reduction in the ecological footprint of shipping is urgently needed in order to contribute to existing sustainable development goals. Within the domain of further reducing air emission levels in container shipping, voluntary declaration and the EEDI are valuable approaches that entail additional measures. The energy consumption approach is valuable to estimate ongoing emission reductions on a continuous basis when no other updated figures are available. The followed approach fills in the current data gap, as the latest worldwide container shipping emissions records date back to 2015.

**Supplementary Materials:** All the datasets originate from the IHS Markit Portal [2] website. The datasets generated and analyzed during the current study are available for purchase from their website at https://maritime.ihs.com/EntitlementPortal/Home/Index.

**Author Contributions:** Conceptualization, E.C.; methodology, E.C. and B.P.; formal analysis, E.C.; investigation, E.C.; resources, E.C., G.T.C., B.P., A.O.-J., T.N.; data curation, E.C., G.T.C.; writing—original draft preparation, E.C., A.O.-J., B.P., G.T.C., T.N.; writing—review and editing, B.P., G.T.C.,

T.N.; visualization and mapping, G.T.C.; supervision, T.N.; funding acquisition, E.C. All authors have read and agreed to the published version of the manuscript.

**Funding:** This research received no external funding.

**Institutional Review Board Statement:** Not applicable.

**Informed Consent Statement:** Not applicable.

**Data Availability Statement:** The datasets generated and analyzed during the study are not publicly available due to IHS Markit Portal [2] copyright but are available from the corresponding author on reasonable request.

**Acknowledgments:** The authors are grateful to the Rector of the University of Gdansk, Professor Piotr Stepnowski as well as to the Dean of the Faculty of Economics, Professor Monika Bąk for supporting and sponsoring the work.

**Conflicts of Interest:** The authors declare no conflict of interest.

## Abbreviations

| | |
|---|---|
| AIS | automatic identification system |
| $CO_2$ | carbon dioxide |
| EEDI | Energy Efficiency Design Index |
| GHG | greenhouse gas |
| GIS | geographic information systems |
| HFO | heavy fuel oil |
| IMO | International Maritime Organization |
| LNG | liquefied natural gas |
| LSHFO | low sulphur heavy fuel oil |
| LSMGO | low sulphur marine gasoil |
| MAE | Med Atlantic Ecobonus |
| MCR | maximum continuous rating |
| MDO | marine diesel oil |
| MGO | marine gasoil |
| $NO_x$ | nitrous oxide |
| PM | particulate matter |
| SCR | selective catalytic reduction |
| SECA | Sulphur Emission Control Area |
| SEEMP | Ship Energy Efficiency Management Plan |
| $SO_x$ | sulphur oxide |
| TEU | twenty-foot equivalent unit |
| VTMS | vessel traffic management system |

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
