# Peer review of "An Energy Consumption Approach to Estimate Air Emission Reductions in Container Shipping"

_energies, doi:10.3390/en14020278_

Round 1
Reviewer 1 Report
Line 21-23 an explanation of abbreviations is only in abstract;
Line 34 “total size” better “tonnage”; see line 38
Line 34 “travel length” better “sailing distance”, “route distance”;
Line 38-40 what the information is for?
Table 1 is bad prepared. It is known 7 types of container vessels: small feeder <1000 TEU, feeder (regional feeder in the article) 1000-2000 TEU, feedermax 2000-3000 TEU, Panamax 3000-5000 TEU, Post-Panamax 5000-10000 TEU, New-Panamax 10000-14500 TEU and ULCS >14501 TEU. Table 1 presents 8 types (without order of capacity!) where I can’t find any New-Panamax vessels!! How is it possible? The presented numbers are in a bad manner.
Line 54 “ship body” better “hull”;
Line 55 “cruise optimization systems” this is a shortage;
Line 79 “state-of-the-art” why not “state of the art”?
Table 2 is bad prepared. The foot-note 1 should be for all in 4-7 lines! In my opinion it should be used [%] not (%) etc. for other units. Below the table there is unclear information!!
Figure 1 is bad prepared!! It was used Polish version of Excel. The diagram should be started from 0.15 to 0.30 on the horizontal axis. There is unit for “g”. Using the Froude Number is possible but better is a dependance between the load of main engine and the vessel speed and next between total fuel consumption and ME power.
Line 186 should be on next page;
Table 3. “Kg/t” better “kg/t”. The numbers of presented energy density are under discussion! A lack information about CmHn and methane slip. That is a big problem to solve it. There is “>0.02” should be “<0.02”. It should be HFO+Scrubber or Scrubbers + SCR in item 9.
Figure 2. MGO is used only for emergency engines! MDO (marine diesel oil) is used as a fuel for other engines and boilers, and so on in farther text.
Line 207 equation (1) – why is so complicated record. Next explanations are still unclear.
Line 229 “180 g/kWh” is OK where the engine load in the range of 50-90% of MCR. Due to slow steaming the engine load is 0.15-0.30 of MCR so the specific fuel consumption is bigger on the level of 185-210 g/kWh!! (depends on the ME power).
Line 243-244 and others “MGO” should be “MDO”;
Line 254-256 I do not fully agree.
Line 272 range of 14-21 kn -as it will be later that is not a true!
Line 723 “max speed at 85 MCR”, vessel speed depends on the type and size and of course the ME nominal power. Calculations taking into account medium speed for all type vessels is a misunderstanding!
Table 4,5,7,8,9,10,11,12,13,14 There is a big mistake or misunderstanding! There is “21 kn” it should be “18 kn”!! Fuel consumption depends in a dependance like in following equation:
FC=a*v3
Please compare. Taking into account (18/17)3 we reach 1.19 ratio of increased fuel consumption where is accordingly 18 and 17 kn. If we take (21/17)3 we reached 1.885 ratio!! The same problem is with the speed “max”.
Table 5 and some others. There is used 6-9 digits. In my opinion the estimation may have a dispersion on a level of 10-50%. Using 9 digits for presented results is a big misunderstanding.
Line 303 “extrapolated on 16 December 2018”? Really, why this day?
Table 9 The Tier III emission is required into ECA areas. The by-pass system is used on outside ECAs. It means about 4 times increased NOx emission (Tier II)! NOx emission depends on the engine load and is not linear dependance with fuel consumption! This is next a big misunderstanding!
Tables present hypotheses! There is no any proposition of predicted container vessel speed in 2020 and next years.
PM emissions from marine diesel oil is low, particularly if a scrubber is used!
Table 13 there is used 10 digits!
Figure 7 is unreadable!
Figure 8 is unreadable! What is “technical efficiency”, what units is used?. What numbers are on the vertical axis? There is only one number 101. Is there the logarithmic scale?
Line 432 - 14-21 kn of course not, but 14-18 kn!
Line 432 – 17 kn is adopted as a reference speed. Why? The speed depends on the type and size of a vessel. A 30 kn speed is possible to achieve for Post-Panamax, New-Panamax and ULCS.
Line 446 and others. Some abbreviations are not explained f.e. LSMGO (of course low sulfur MGO).
Line 459 “cleaning of fuel injectors” – how to do it?, “correct fuel injector viscosity” rather “fuel viscosity before injectors” etc. There are many problems with proper specialist terminology!
Conclusions – under discussion with the professionals!
Author Response
We would like to thank the reviewer for the valuable comments which allowed us to improve the manuscript. As stated in the manuscript and the conclusions, we have tried to present a model that can generate a generalized view on emission volumes of the global container fleet. Given the thousands of ships and the many trade routes on which they operate, the model includes some assumptions which might simplify reality (all models do so), but make the calculations feasible and less data intensive. By doing so, we believe we were able to present good results which have relevance to policy makers and practitioners.

Reviewer 2 Report
I want to thank the authors for the opportunity to read this interesting paper. Environmental sustainability is always a welcomed subject, and researchers should strive to find a means to protect the environment and prevent pollution. This paper aims to use an energy consumption approach to estimate air emission reductions in container shipping. The authors presented their research very nicely, adequately described methods, and presented their results clearly.
Nevertheless, I have just a few minor concerns regarding this paper given in the comments below.
- In Table 1, the authors lined up types of container ships alphabetically. However, I propose that authors line up container vessels according to size, which is the more common and more logical lineup.
- Line 54: "cruise optimisation systems" are mentioned. I suggest changing the wording to: "voyage optimization systems" because it is a more common phrase in maritime transport.
- Line 156: the wording "waiting at the quay" is used, and that could be misleading since vessels usually perform cargo operations and spend a short time in idle positions. I suggest changing the wording.
- Line 187: "Selective catalytic reaction" is mentioned, but shouldn't it be "Selective catalytic reduction"?
- Line 248: it is stated that the voyage consists of four stages, and one stage is "mooring or berthing (i.e., handling operations at the quay). I propose to change wording since the current construct can be misleading. Mooring or berthing is a part of a maneuvering stage since the vessel is still being moored at the berth and handling mooring lines. Furthermore, the text in the parenthesis did not specify what kind of handling is being performed. Authors could use the wording "moored at berth (or a quay) for cargo handling operations," giving a clearer picture to the reader about operations carried during the port stay.
- Line 250: "Time at sea (i.e., cruising)" is mentioned. I suggest changing the word "cruising" into "sailing" because it's meaning suits better in this context.
- In Figure 3, word maneuvering is incorrectly typed ("Manuevring").
- In Figure 3, the word "cruising" is used. I suggest the same as under 6.
- Line 516: "Selective catalytic reaction" is mentioned, but shouldn't it be "Selective catalytic reduction"?
The paper's aim was reached, and the authors used an energy consumption approach and estimated air emission reductions in container shipping.
Thank you for the opportunity to review your paper, and I hope that my comments will be useful to improve it.
Author Response
We would like to thank the reviewer for the valuable comments which allowed us to improve the manuscript.

Round 2
Reviewer 1 Report
Only emissions of CO2 and SOx are approximately linear dependence to fuel consumption. NOx and PM emissions depend on the load of engines but as approximation I may agree with the statements.